Report

# The dual role of Spn-E in supporting heterotypic ping-pong piRNA amplification in silkworms

Natsuko Izumi [ID][1,5], Keisuke Shoji [ID][1,2,3,5], Lumi Negishi[4] & Yukihide Tomari [ID][1,2 ✉]

## Abstract

The PIWI-interacting RNA (piRNA) pathway plays a crucial role in silencing transposons in the germline. piRNA-guided target cleavage by PIWI proteins triggers the biogenesis of new piRNAs from the cleaved RNA fragments. This process, known as the ping-pong cycle, is mediated by the two PIWI proteins, Siwi and BmAgo3, in silkworms. However, the detailed molecular mechanism of the ping-pong cycle remains largely unclear. Here, we show that Spindle-E (Spn-E), a putative ATP-dependent RNA helicase, is essential for BmAgo3-dependent production of Siwi-bound piRNAs in the ping-pong cycle and that this function of Spn-E requires its ATPase activity. Moreover, Spn-E acts to suppress homotypic Siwi–Siwi ping-pong, but this function of Spn-E is independent of its ATPase activity. These results highlight the dual role of Spn-E in facilitating proper heterotypic ping-pong in silkworms.

**Keywords** BmAgo3; Ping-Pong Cycle; piRNA; Siwi; Spn-E
**Subject Category** RNA Biology

## Introduction

The PIWI-interacting RNA (piRNA) pathway is a widely conserved RNA silencing mechanism against transposable elements (TEs) in the animal germline (Ozata et al, 2019; Wang et al, 2023). piRNAs are typically 24–31 nt single-stranded RNAs that guide their associated PIWI proteins to complementary target RNAs. While some PIWI proteins induce transcriptional repression in the nucleus (Aravin et al, 2008; Kuramochi-Miyagawa et al, 2008; Sienski et al, 2012), many PIWI proteins function as endoribonucleases and inhibit target expression by endonucleolytic cleavage in the cytoplasm (Brennecke et al, 2007; Gunawardane et al, 2007; De Fazio et al, 2011; Reuter et al, 2011).

PIWI-catalyzed target cleavage not only suppresses the target but also produces new piRNAs. In this process, known as the ping-pong cycle, the 3′ fragment of PIWI-cleaved target RNA is incorporated into another PIWI as a new piRNA precursor (pre-pre-piRNA) (Ozata et al, 2019; Wang et al, 2023). The PIWI-loaded pre-pre-piRNA undergoes additional cleavage at a downstream position by another PIWI protein or the endonuclease Zucchini (Zuc, MitoPLD in mice) to become a precursor piRNA (pre-piRNA) (Ozata et al, 2019; Gainetdinov et al, 2018; Izumi et al, 2020). The cleavage by Zuc not only contributes to the pre-piRNA production but also generates the 5′ end of a new pre-pre-piRNA from the downstream cleavage fragment, expanding the production of trailing piRNAs to the 3′ direction (Mohn et al, 2015; Han et al, 2015; Ozata et al, 2019; Gainetdinov et al, 2018). In many species including silkworms, the 3′ end of pre-piRNAs is further trimmed by the exonuclease Trimmer (PNLDC1 in mice) and 2′-O-methylated by Hen1 (HENMT1 in mice) (Nishimura et al, 2018; Ding et al, 2017; Izumi et al, 2016; Zhang et al, 2017; Horwich et al, 2007; Saito et al, 2007; Kirino and Mourelatos, 2007). The resulting mature piRNA, in turn, guides its associated PIWI protein to cleave a complementary target RNA, thereby producing a new piRNA precursor from the opposite strand. Because PIWI proteins cleave target RNAs precisely at a position between the 10th and 11th nucleotides of the guiding piRNA, the ping-pong cycle amplifies pairs of piRNAs that have a 10-nucleotide complementary sequence at their 5′ ends (Brennecke et al, 2007; Gunawardane et al, 2007; Aravin et al, 2008). A subset of PIWI proteins preferentially binds to piRNAs with uridine at the first nucleotide (1U) (Wang et al, 2023), and they produce piRNAs with an adenine bias at the 10th position (10 A) by ping-pong amplification due to the nucleotide preference of the PIWI proteins in target cleavage (Wang et al, 2014). In silkworms, the ping-pong cycle typically occurs between the two PIWI proteins, Siwi and BmAgo3 (Kawaoka et al, 2009; Nishida et al, 2015). Siwi-bound piRNAs have a 1U and antisense bias, while BmAgo3-bound piRNAs have a 10A and sense bias (Kawaoka et al, 2009; Nishida et al, 2015).

Small RNA-guided target cleavage is a common mode of action for the AGO and PIWI clades of Argonaute (Ago) family proteins (Hutvagner and Simard, 2008; Ozata et al, 2019). Unlike AGO proteins, however, PIWI proteins are thought to require ATP-dependent helicases for the efficient release of cleaved RNA fragments (Nishida et al, 2015; Murakami et al, 2021). So far, two ATP-dependent DEAD-box RNA helicases, Vasa and DDX43, have been proposed as the factors responsible for releasing cleavage fragments from Siwi and BmAgo3, respectively (Nishida et al, 2015; Murakami et al, 2021). Vasa-EQ, a DEAD-box mutant of Vasa,

[1]Laboratory of RNA Function, Institute for Quantitative Biosciences, The University of Tokyo, Bunkyo-ku, Tokyo 113-0032, Japan. [2]Department of Computational Biology and Medical Sciences, Graduate School of Frontier Sciences, The University of Tokyo, Bunkyo-ku, Tokyo 113-0032, Japan. [3]Graduate School of Bio-Applications and Systems Engineering, Tokyo University of Agriculture and Technology, Koganei-shi, Tokyo 184-8588, Japan. [4]Laboratory of Chromatin Structure and Function, Institute for Quantitative Biosciences, The University of Tokyo, Bunkyo-ku, Tokyo 113-0032, Japan. [5]These authors contributed equally: Natsuko Izumi, Keisuke Shoji. ✉E-mail: tomari@iqb.u-tokyo.ac.jp

lacks the ability to release ATP hydrolysis products and therefore freezes the Siwi-to-BmAgo3 ping-pong intermediate complex (called "Amplifier"), which contains Vasa, piRNA-bound Siwi, BmAgo3, and Qin (Xiol et al, 2014).

piRNA pathway components often exhibit distinct patterns of cellular localization. For example, PIWI proteins and many piRNA factors (e.g., BmAgo3 and Vasa) accumulate in phase-separated perinuclear granules called "nuage" (Lim and Kai, 2007; Aravin et al, 2009; Nishida et al, 2015), while other piRNA factors (e.g., Spindle-E [Spn-E] and Qin) localize to another type of cytoplasmic granule known as processing bodies (P-bodies), where RNA-degradation enzymes are enriched (Aravin et al, 2009; Chung et al, 2021; Standart and Weil, 2018). In addition, several piRNA processing factors (e.g., Zuc and Trimmer) are located on the mitochondrial surface through their transmembrane domains (Choi et al, 2006; Saito et al, 2010; Patil et al, 2017; Izumi et al, 2016). Their dynamic subcellular compartmentalization and coordinated actions are critical for piRNA biogenesis (Ge et al, 2019; Chung et al, 2021).

Spn-E, a DExH-box RNA helicase that also features a Tudor domain (Gillespie and Berg, 1995; Siomi et al, 2010), is a conserved piRNA factor essential for piRNA-mediated TE silencing (Aravin et al, 2001; Vagin et al, 2006; Shoji et al, 2009; Wenda et al, 2017; Chen et al, 2023). In contrast to its mouse ortholog Tdrd9, which only modestly affects the piRNA expression profile (Shoji et al, 2009; Wenda et al, 2017), Spn-E is essential for piRNA biogenesis in flies and silkworms (Vagin et al, 2006; Lim and Kai, 2007; Malone et al, 2009; Nishida et al, 2015). In both flies and silkworms, mutations in the ATPase domain of Spn-E fail to support normal piRNA biogenesis and TE silencing (Ott et al, 2014; Nishida et al, 2015), indicating that ATPase activity is crucial for Spn-E function. Moreover, in spn-E mutant flies, Aub and Ago3 are mislocalized from nuage, and the production of both sense and antisense germline piRNAs is severely compromised (Lim and Kai, 2007; Malone et al, 2009), suggesting a defect in the ping-pong cycle. Silkworm Spn-E forms a complex that contains Siwi and Qin but lacks BmAgo3 and Vasa, and the depletion of Spn-E causes a reduction in both Siwi- and BmAgo3-bound piRNAs (Nishida et al, 2015). Accordingly, a model has been proposed in which Spn-E acts in the production of Siwi-bound "primary" piRNAs, under the assumption that the biogenesis of BmAgo3-bound "secondary" piRNAs depends on the target cleavage by Siwi-bound "primary" piRNAs via ping-pong (Nishida et al, 2015). However, the discovery that target cleavage in ping-pong in fact triggers the production of trailing piRNAs has challenged the original definition of "primary" and "secondary" piRNAs (Mohn et al, 2015; Han et al, 2015; Ozata et al, 2019), thereby blurring the specific role of Spn-E in the piRNA biogenesis pathway.

Here, we investigated the role of Spn-E using an ATPase-deficient mutant, Spn-E-EQ (E251Q), in BmN4 cells derived from silkworm ovaries. We observed that Spn-E-EQ forms aggregates distinct from P-bodies together with BmAgo3 and increases the precursors of Siwi-bound piRNAs. Spn-E knockdown (KD) primarily reduced Siwi-bound piRNAs generated by BmAgo3-mediated target cleavage. Wild-type Spn-E, but not the EQ mutant, could partially restore the reduction in BmAgo3-bound piRNAs, suggesting the requirement of the ATPase activity for this function of Spn-E. Unexpectedly, we also found that Spn-E KD enhances Siwi–Siwi homotypic ping-pong independently of its ATPase activity. The necessity of Spn-E for BmAgo3-dependent Siwi-bound piRNA production was further supported by artificial piRNA reporter experiments in BmN4 cells. In contrast, we found

no evidence to support the requirement of DDX43 for piRNA biogenesis in cells, even though the recombinant DDX43 protein showed a robust activity to release the cleavage products of BmAgo3 in vitro. Our results suggest an essential role for Spn-E in facilitating the production of Siwi-bound piRNAs in the canonical heterotypic ping-pong cycle.

# Results and discussion

## Siwi knockdown accumulates Spn-E in BmAgo3 complexes

In the ping-pong cycle of silkworms, the 3′ cleavage fragments produced by BmAgo3 are loaded into Siwi as a pre-pre-piRNA and vice versa (Kawaoka et al, 2009; Nishida et al, 2015). When we analyzed immunopurified BmAgo3 complexes from the lysate of Siwi-KD BmN4 cells, we observed a remarkable accumulation of two proteins, p160 and p40 (Fig. 1A). LC-MS/MS analysis identified p160 as Spn-E, one of the conserved piRNA factors (p40 will be described elsewhere (Izumi et al, manuscript in preparation)). In line with the increased physical interaction, Siwi KD caused an accumulation of Spn-E in BmAgo3-containing nuage in BmN4 cells (Fig. 1B). In theory, the depletion of Siwi should lead to an accumulation of BmAgo3, which remains bound to its cleavage fragments that cannot be handed over to Siwi (Nishida et al, 2020). Accordingly, we speculated that Spn-E functions during the handover process from BmAgo3 to Siwi in the ping-pong cycle and that Siwi KD causes Spn-E to stall at an intermediate step in this process.

## ATPase-deficient Spn-E-EQ exhibits increased association with BmAgo3

Spn-E encodes a putative ATP-dependent RNA helicase belonging to the DExH-box family (Gillespie and Berg, 1995), and its ATPase activity is essential for the production of Siwi-bound piRNAs in silkworms (Nishida et al, 2015). To determine the process that requires the ATPase activity of Spn-E, we examined the behavior of the ATPase-deficient mutant Spn-E-EQ (E251Q). We previously reported that Spn-E localizes primarily to P-bodies (Chung et al, 2021), while a fraction of Spn-E partially localizes to BmAgo3-containing nuage (Fig. 1B). To minimize the effect of endogenous Spn-E, we depleted it using RNAi with double-stranded RNAs (dsRNAs) targeting the Spn-E 3′ UTR, and complemented the cells with either wild-type Spn-E or the EQ mutant. Wild-type Spn-E exhibited a dotted distribution throughout the cytoplasm, with partial colocalization with BmAgo3-containing nuage in the perinuclear region (Fig. 1C). In contrast, Spn-E-EQ formed aberrant aggregates with BmAgo3 (Fig. 1C), which was reminiscent of the aggregates formed by Vasa-EQ with BmAgo3 as previously reported (Xiol et al, 2014). These results suggest that Spn-E shows increased association with BmAgo3 in Siwi KD or Spn-E-EQ expression. To confirm this, we immunoprecipitated Spn-E-EQ and examined co-precipitated Siwi and BmAgo3. As expected, the EQ mutation of Spn-E reproducibly enhanced its association with BmAgo3 more than that with Siwi, although the p-values for the enhancing effects were slightly above 0.05 by t-test with Holm correction (Figs. 1D and EV1A). Because Spn-E also localizes to

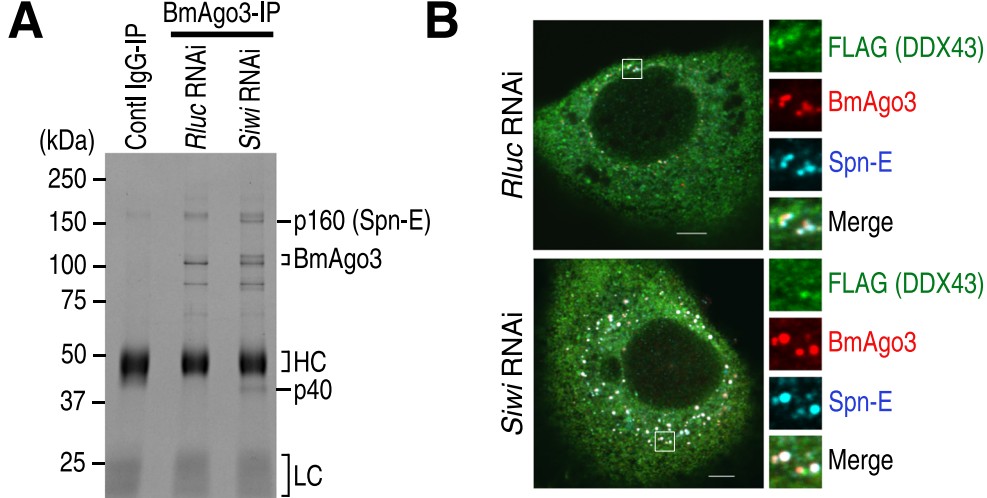

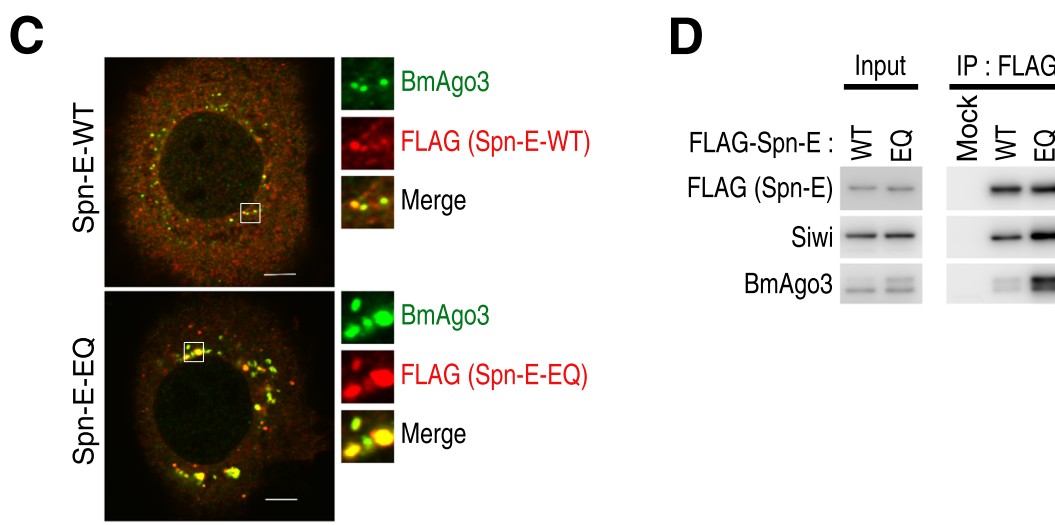

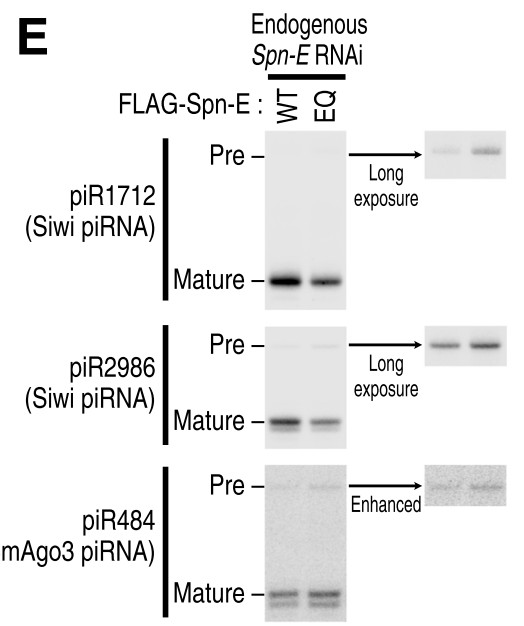

**Figure 1.  The ATPase-deficient Spn-E-EQ forms aggregates with BmAgo3.**

(A) CBB staining of immunoprecipitated BmAgo3 complexes from BmN4 cells treated with dsRNA targeting *Rluc* or *Siwi*. *Rluc*; *Renilla luciferase*, control. HC, IgG heavy chain; LC, IgG light chain. (B) Subcellular localization of BmAgo3, Spn-E, and FLAG-DDX43 in BmN4 cells treated with dsRNA targeting *Rluc* or *Siwi*. *Rluc*; *Renilla luciferase*, control. Scale bar, 5 μm. (C) Subcellular localization of FLAG-Spn-E (wild type or EQ) and BmAgo3 in BmN4 cells treated with dsRNA targeting the *Spn-E* 3′ UTR. Scale bar, 5 μm. (D) Western blot analysis of immunopurified FLAG-Spn-E (wild-type or EQ) complexes from BmN4 cells treated with dsRNA targeting the *Spn-E* 3′ UTR. Quantification data from three independent experiments are shown in Fig. EV1A. (E) Northern blot analysis of piR1712, piR2986, and piR484 in BmN4 cells co-transfected with the FLAG-Spn-E (wild type or EQ) plasmid and dsRNA targeting the *Spn-E* 3′ UTR. Quantification data for mature piRNA and pre-piRNA signals from four independent experiments are shown in Fig. EV1F,G, respectively. Source data are available online for this figure.

P-bodies (Chung et al, 2021), we investigated the localization of DDX6, a marker protein for P-bodies, in cells expressing Spn-E-EQ using an anti-DDX6 antibody (Fig. EV1B). Although DDX6 was occasionally found near the aggregates of Spn-E-EQ and BmAgo3, it did not overlap with these proteins (Fig. EV1C). Thus, the Spn-E-EQ-BmAgo3 aggregates are probably not P-bodies themselves, but a fraction of them may fuse with P-bodies as these aggregates grow. We also examined the localization of Siwi in Spn-E-EQ-expressing cells. We found that a fraction of Siwi colocalized with Spn-E-EQ-BmAgo3 aggregates but to a lesser extent compared to BmAgo3 (Fig. EV1D). In addition, we observed some Siwi aggregates without Spn-E and BmAgo3 (Fig. EV1D). These results suggest that the ATPase-deficient Spn-E-EQ forms an aberrant complex that is stuck with BmAgo3 and possibly with Siwi as well.

## Spn-E-EQ reduces mature Siwi-bound piRNAs while accumulating their pre-piRNAs

To examine whether both Siwi and BmAgo3 are contained within the same Spn-E-EQ complex, we performed a tandem IP experiment. We first immunoprecipitated FLAG-tagged Spn-E-EQ, and the purified Spn-E-EQ complex was subsequently subjected to a second immunoprecipitation with an anti-BmAgo3 antibody to detect the co-precipitated Siwi in the presence or absence of RNase treatment. Siwi was co-purified with BmAgo3 in the second IP, and the level was largely unaffected by RNase treatment (Fig. EV1E). These results suggest that Spn-E-EQ, BmAgo3, and Siwi form an aberrantly stable complex. Given that both Siwi and BmAgo3 are co-immunoprecipitated with wild-type Spn-E, albeit less strongly than Spn-E-EQ (Figs. 1D and EV1A), we speculate that this tertiary complex is formed at least transiently even under normal conditions.

If Spn-E uses its ATPase activity to facilitate the handover process from BmAgo3 to Siwi, the expression of Spn-E-EQ should, in theory, cause an accumulation of RNA fragments cleaved by BmAgo3, which would become pre-pre-piRNAs for Siwi. To explore this possibility, we attempted to detect piR1712 and piR2986, two representative Siwi-bound piRNAs generated by BmAgo3-mediated cleavage, and their precursors in BmN4 cells expressing Spn-E-EQ by northern blotting. Compared to wild-type Spn-E, Spn-E-EQ caused a reduction in mature piR1712 and piR2986 and a concomitant accumulation of longer RNA signals corresponding to the lengths of their pre-piRNAs (Figs. 1E and EV1F,G). We also observed slight accumulation of pre-piR484, the precursor of a BmAgo3-bound piRNA, by expressing Spn-E-EQ, but an accompanying reduction in mature piR484 was not observed (Figs. 1E and EV1F,G). In general, pre-pre-piRNAs are cleaved at a downstream position either by Zuc or another piRNA-guided PIWI protein to become pre-piRNAs (Izumi et al, 2020).

Unlike Zuc-mediated cleavage, downstream cleavage by PIWI proteins can occur before pre-pre-piRNAs are loaded into new PIWI proteins. After PIWI loading, pre-piRNAs are rapidly trimmed to their mature length, making them undetectable (Izumi et al, 2020). Therefore, the pre-piRNAs detected here are likely the cleavage products of downstream PIWI proteins prior to being loaded into new PIWI proteins. Taken together, we concluded that Spn-E-EQ inhibits the handover of BmAgo3-cleaved fragments to Siwi.

## Spn-E does not have an activity to release Ago3-mediated cleavage fragments

As Spn-E encodes an RNA helicase, it is conceivable that Spn-E has a function to dissociate the cleavage products from BmAgo3 to facilitate their handover to Siwi. However, DDX43, another DEAD-box RNA helicase, has been reported to be the factor responsible for releasing the cleavage fragments of BmAgo3 (Murakami et al, 2021). To determine if Spn-E exhibits activity similar to DDX43, we repeated the previously reported in vitro assay to examine the release of the cleavage fragments from BmAgo3. To distinguish the 5′ and 3′ cleavage fragments, we prepared a target RNA radiolabeled with $^{32}$P at different positions and performed an in vitro target cleavage reaction using immunoprecipitated BmAgo3. After that, we added recombinant Spn-E protein (rSpn-E) or DDX43 protein (rDDX43) to the reaction and monitored the release of the cleavage fragments in the supernatant (Fig. EV1H,I). Consistent with the previous report (Murakami et al, 2021), rDDX43 released both the 5′ and 3′ cleavage fragments into the supernatant (Fig. EV1I). However, despite possessing the ATPase activity (Fig. EV1J), rSpn-E did not release the cleavage fragments from BmAgo3 (Fig. EV1I). Thus, unlike DDX43, Spn-E does not have the releasing activity for BmAgo3-cleaved RNA fragments in vitro.

To compare the effect of DDX43 dysfunction and Spn-E dysfunction on the cellular status of BmAgo3, we expressed the previously reported ATPase-deficient mutant of DDX43, DDX43-D399A (DA), which retains the capacity to bind BmAgo3 (Murakami et al, 2021), in BmN4 cells and examined the localization of BmAgo3. In agreement with the previous report (Murakami et al, 2021), DDX43 showed dispersed localization patterns in the cytoplasm, with only occasional and partial overlap with BmAgo3-containing nuage (Figs. 1B and EV1K). This pattern was also observed with DDX43-DA (Fig. EV1K). Unlike Spn-E-EQ, DDX43-DA neither formed aggregates with BmAgo3 nor affected the subcellular distribution of BmAgo3 (Fig. EV1K). In addition, the localization pattern of DDX43 was unaffected by either KD of Siwi or expression of Spn-E-EQ (Figs. 1B and EV1L). Overall, DDX43 had much less impact on BmAgo3 in BmN4 cells compared to Spn-E.

## Depletion of Spn-E decreases the BmAgo3-dependent production of Siwi-bound piRNAs and increases the Siwi–Siwi homotypic ping-pong

We next knocked down Spn-E or DDX43 using two different dsRNAs, respectively (Fig. EV2A,B) and examined the impact of KD on piRNA expression. We first analyzed the changes in the expression of piRNAs derived from TEs. While most TEs showed decreased piRNA production in Spn-E KD, a subset of TEs unexpectedly exhibited increased piRNA production (Fig. EV2C, top and middle, red dots). In contrast, DDX43 KD caused little or no change in the production of TE-derived piRNAs (Fig. EV2C, bottom). Notably, Murakami et al, also failed to detect any apparent effects on endogenous piRNAs upon DDX43 KD (Murakami et al, 2021). To determine the identity of the increased piRNAs in Spn-E KD, we classified the TE-mapped piRNAs into three groups according to their changes in expression and examined their nucleotide bias for 1U and 10A, which are hallmarks of Siwi- and BmAgo3-bound piRNAs, respectively (Figs. 2A,B and EV2D−F). We found that over half of the decreased piRNAs in Spn-E KD are "1U but not 10A" piRNAs (Figs. 2B [RNAi targeting 3′ UTR] and EV2F [RNAi targeting CDS]). Moreover, our re-analysis of previously reported Siwi-IP and BmAgo3-IP piRNA libraries revealed that, as expected, these "1U but not 10A" piRNA species are predominantly bound to Siwi and that their putative partner piRNAs in the ping-pong cycle (i.e., complementary piRNAs bearing a 5′ 10-nt overlap) are predominantly bound to BmAgo3 in normal BmN4 cells (Fig. EV2G, top). Therefore, Spn-E is required for the production of Siwi-bound piRNAs via BmAgo3-mediated target cleavage. On the other hand, most of the increased piRNAs in Spn-E KD had the "1U and 10A" bias (Figs. 2B and EV2F). Importantly, the "1U and 10A" piRNAs and their partner piRNAs with 5′ 10-nt complementarity were both concentrated in the Siwi-IP (Fig. EV2G, bottom), indicating that they are largely produced through a homotypic ping-pong between Siwi–Siwi.

Based on the above observation, we again categorized the TE-mapped piRNAs into three different types: A → S, S → A, and S → S piRNAs (Fig. 2C), where "S" and "A" represent Siwi and BmAgo3, respectively, and the arrow indicates the flow of RNA fragments during ping-pong. For example, A → S piRNAs are predominantly bound to Siwi and have partner piRNAs with 5′ 10-nt complementarity that are predominantly bound to BmAgo3 (i.e., they are Siwi-bound piRNAs produced via BmAgo3-mediated target cleavage). On the other hand, S → S piRNAs are predominantly bound to Siwi and are generated mainly through the Siwi–Siwi homotypic ping-pong. We analyzed the change in the relative abundance of each piRNA type in Spn-E KD or DDX43 KD. In agreement with the analysis based on the 1U and 10 A biases (Figs. 2B and EV2F), A → S piRNAs were decreased, and S → S piRNAs were conversely increased by Spn-E KD (Fig. 2D). In contrast, DDX43 KD had no noticeable effect on the relative abundance of any of the groups (Fig. EV2H). Thus, Spn-E, but not DDX43, is required for the BmAgo3-dependent production of Siwi-bound piRNAs and for the suppression of Siwi–Siwi homotypic ping-pong.

We next examined the requirement of the ATPase activity of Spn-E for the production of A → S piRNAs and suppression of S → S piRNAs. We constructed small RNA libraries from BmN4 cells, where endogenous Spn-E was knocked down, and either wild-type Spn-E or the ATPase-deficient mutant Spn-E-EQ was expressed. Wild-type Spn-E but not the EQ mutant partially recovered the decrease in A → S piRNAs (Fig. 2E), suggesting that the ATPase activity of Spn-E is required for A → S piRNA production (Fig. 2F). On the other hand, the enhanced production of S → S piRNAs was rescued by the expression of both wild-type Spn-E and the EQ mutant (Fig. 2E), indicating that the presence of Spn-E itself, rather than its ATPase activity, is important for repressing the Siwi–Siwi homotypic ping-pong (Fig. 2F).

## Spn-E but not DDX43 is required for artificial A→S piRNA production

To confirm the requirement of Spn-E for de novo A → S piRNA production, we developed a reporter system that generates an artificial Siwi-bound piRNA, piR484-A (Figs. 3A and EV3A). This reporter RNA has a target site for an abundantly expressed endogenous BmAgo3-bound piRNA and the resulting cleavage product is expected to be loaded into Siwi via the ping-pong cycle. It also contains another downstream target site for an endogenous Siwi-bound piRNA to define the formation of the 3′ end of the pre-piRNA; following cleavage at this downstream position by Siwi, 3′ end maturation by Trimmer and Hen1 is expected to occur. We co-transfected this reporter plasmid and dsRNAs targeting Spn-E or DDX43 into BmN4 cells, and detected Siwi-bound piR484-A by northern blotting. As expected, the production of piR484-A was reduced by Spn-E KD (Fig. 3B). In contrast, DDX43 KD did not affect the production of piR484-A (Fig. 3B), even though the DDX43 mRNA level was markedly decreased by KD (Fig. EV3B).

We also tested another reporter that has a target site for a piggyBac transposon-derived piRNA, which is essentially the same as the artificial Siwi-bound piRNA reporter used in the previous study (Murakami et al, 2021), except that it lacks the upstream EGFP sequence (Fig. 3C). Unlike the piR484-A reporter, it lacks any specific downstream piRNA target site that can define the 3′ end of the pre-piRNA. To ensure that the reporter piRNA is generated even without 3′ end processing, we used not only the circular plasmid but also a linearized plasmid that produces a run-off transcript with a defined 3′ end. In theory, the reporter RNA transcribed from the circular plasmid is expected to have a >60 nt sequence and a poly(A) tail downstream of the cleavage site, so the exact mechanism for 3′ end processing after Siwi-loading is unclear (Fig. 3C, left). On the other hand, the BmAgo3-cleavage product of the reporter RNA transcribed from the linearized plasmid should yield a 30-nt fragment, which is expected to be loaded into Siwi as an artificial piRNA without requiring additional 3′ end processing (Fig. 3C, right). Similar to the result from the piR484-A system, the production of the artificial Siwi-bound piRNA was reduced by Spn-E KD but not by DDX43 KD, for both circular and linearized plasmids (Figs. 3D and EV3B). These results strongly support the requirement of Spn-E for A → S piRNA production.

To confirm the necessity of Spn-E ATPase activity for artificial Siwi-bound piRNA production, we performed a KD rescue experiment. We co-transfected the piggyBac piRNA reporter plasmid and either the wild-type Spn-E or the EQ mutant-expressing plasmid, under the knockdown of endogenous Spn-E. Consistent with the analysis of endogenous piRNA expression (Fig. 2E), wild-type Spn-E but not the EQ mutant rescued the decrease in the artificial Siwi-bound piRNA caused by Spn-E KD (Fig. 3E). Similar results were obtained with the piR484-A reporter

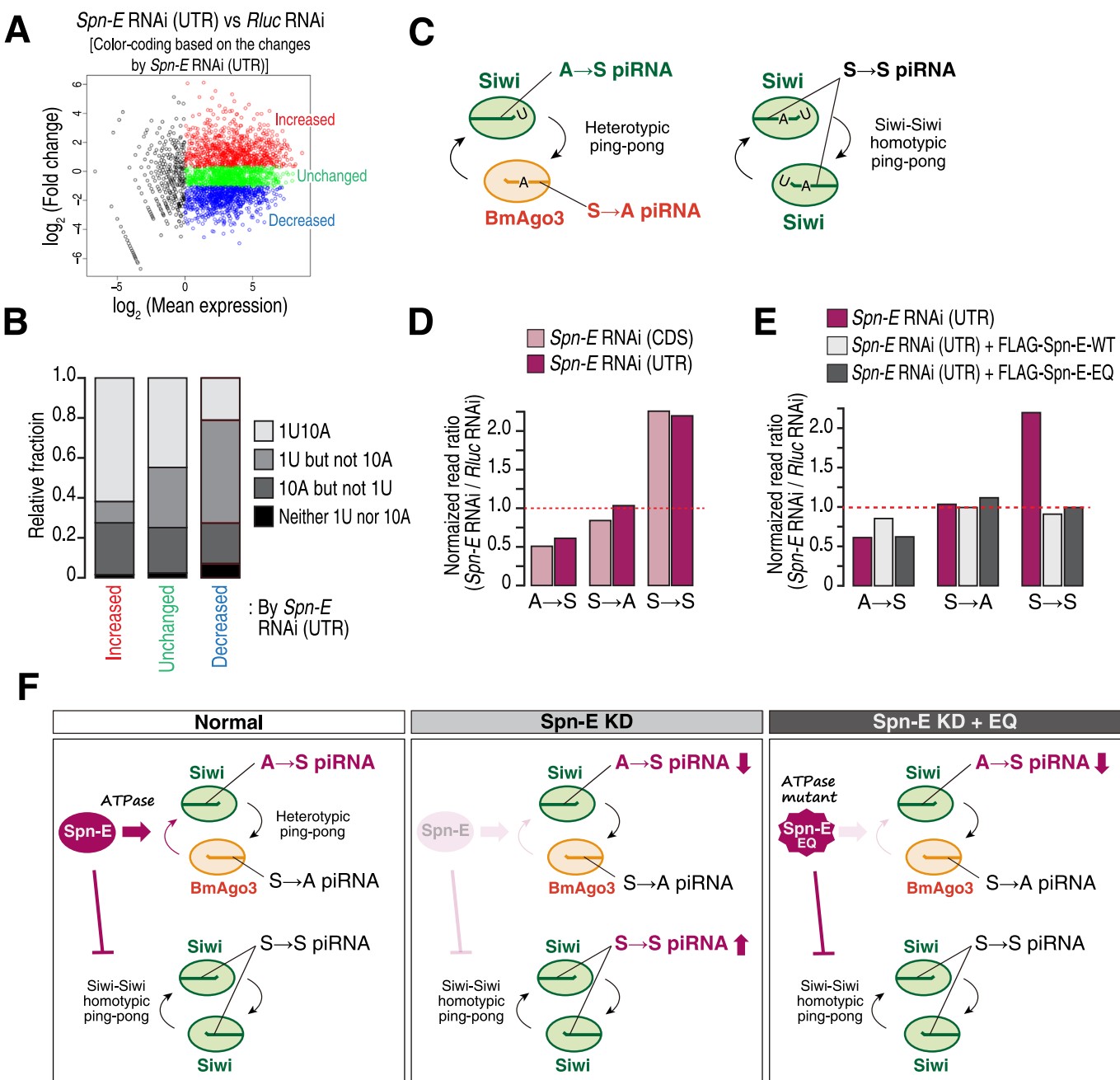

**Figure 2. Depletion of Spn-E decreases BmAgo3-dependent production of Siwi-bound piRNAs while increasing Siwi–Siwi homotypic ping-pong.**

(A) MA plot showing piRNA expression changes between control KD (*Rluc* RNAi) and *Spn-E* KD using dsRNA targeting the *Spn-E* 3′ UTR. Each dot represents one piRNA. Based on the changes in expression, piRNAs were divided into three groups: "increased" (red, *n* = 825), "unchanged" (green, *n* = 824), and "decreased" (blue, *n* = 825). (B) Relative fractions of 1U10A, 1U but not 10A, 10A but not 1U, and neither 1U nor 10A piRNAs of each group defined in (A). (C) Schematic representation of A → S, S → A, and S → S piRNAs. (D) Changes in the expression of A → S, S → A, and S → S piRNAs in *Spn-E* KD relative to control KD (*Rluc* RNAi). Two different dsRNAs targeting the *Spn-E* CDS and 3′ UTR were used. (E) Changes in the expression of A → S, S → A, and S → S piRNAs in the indicated conditions relative to control KD (*Rluc* RNAi). The *Spn-E* RNAi (UTR) data in (D) are included for comparison. (F) Summary of the results from the *Spn-E* KD rescue experiment. *Spn-E* KD decreases A → S piRNAs and increases S → S piRNAs. A → S piRNA production requires the ATPase activity of Spn-E, whereas the suppression of S → S piRNA production does not. Source data are available online for this figure.

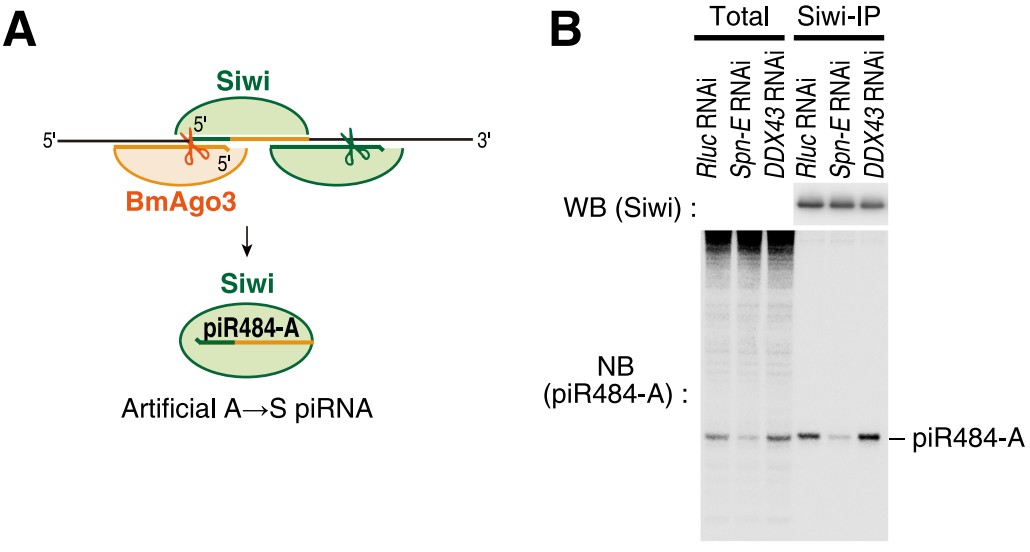

**A**

Siwi

BmAgo3

Siwi
piR484-A

Artificial A→S piRNA

**B**

| | Total | Siwi-IP |
|---|---|---|

WB (Siwi) :

NB
(piR484-A) :

— piR484-A

**C**

Circular plasmid

Linearized plasmid

No GFP
(No ORF)

*piggyBac* piRNA target seq

5'                    (A)n
BmAgo3        > 60 nt

5'
BmAgo3    20 nt

Siwi            (A)n
           ✂ ?
       Trimmer

Siwi

30 nt
(Expected Siwi piRNA)

**D**

| Circular | | Linearized | |
|---|---|---|---|
| Total | Siwi-IP | Total | Siwi-IP |

WB (Siwi) :

NB
(art-Siwi piRNA) :

— art-Siwi
   piRNA

**E**

Circular *piggyBac*
piRNA reporter

WB

Input

FLAG

Spn-E

Tubulin

Siwi-IP

Siwi

NB

art-Siwi
piRNA

**Figure 3. Artificial Siwi-bound piRNA production is impaired by Spn-E KD but not by DDX43 KD.**

(A) Schematic representation of the piR484-A reporter. piR484-A is an artificial A → S piRNA with a chimeric sequence of piR484 and piR2986. The details are shown in Fig. EV3A. (B) Siwi was immunoprecipitated from BmN4 cells co-transfected with the piR484-A reporter plasmid and dsRNA targeting the indicated genes for RNAi. Immunoprecipitated Siwi and Siwi-bound piR484-A were detected by western blotting and northern blotting, respectively. *Rluc; Renilla luciferase*, control. (C) Schematic representation of the production of *piggyBac* piRNA reporter-derived artificial Siwi-bound piRNAs. Left: The circular plasmid produces a reporter RNA with a long 3′ region. After cleavage by *piggyBac* piRNA-loaded BmAgo3, the 3′ end of the Siwi-loaded reporter RNA is processed by an unknown downstream cleavage event and/or trimming by Trimmer. Right: After cleavage by *piggyBac* piRNA-loaded BmAgo3, the reporter RNA from the linearized plasmid is expected to be 30-nt long, which is similar in length to that of endogenous piRNAs. Consequently, additional 3′ end processing is unnecessary in theory. (D) Siwi was immunoprecipitated from BmN4 cells co-transfected with a circular or linearized *piggyBac* piRNA reporter plasmid and dsRNA targeting the indicated genes for RNAi. Immunoprecipitated Siwi and reporter-derived artificial Siwi-bound piRNAs (art-Siwi piRNAs) were detected by western blotting and northern blotting, respectively. *Rluc; Renilla luciferase*, control. (E) KD rescue experiment of Spn-E using the *piggyBac* piRNA reporter. Siwi was immunoprecipitated from BmN4 cells co-transfected with a circular *piggyBac* piRNA reporter plasmid, the FLAG-Spn-E expression plasmid, and dsRNA targeting the *Spn-E* 3′ UTR. Immunoprecipitated Siwi and reporter-derived artificial Siwi-bound piRNAs (art-Siwi piRNAs) were detected by western blotting and northern blotting, respectively. *Rluc; Renilla luciferase*, control. Source data are available online for this figure.

(Fig. EV3C). Taken together, we concluded that the ATPase activity of Spn-E is required for BmAgo3-dependent production of Siwi-bound piRNAs during the ping-pong cycle.

Although Spn-E has long been studied as a conserved piRNA factor, its role in piRNA biogenesis has remained unclear. In this study, we showed that Spn-E is required for the BmAgo3-mediated production of Siwi-bound piRNAs in silkworms. The notable reduction in A → S piRNAs in Spn-E KD (Fig. 2D) and the phenotype observed in the EQ mutant (Fig. 1) strongly suggest the role of Spn-E in the ping-pong cycle. Since the expression of Spn-E-EQ increases A → S pre-piRNAs (Figs. 1E and EV1G), Spn-E is expected to act as an ATPase after target cleavage by BmAgo3 before the handover of the cleaved fragment to Siwi. However, as purified Spn-E did not exhibit an activity to release BmAgo3-mediated cleavage fragments in vitro (Fig. EV1I), Spn-E is unlikely to directly dissociate the cleaved fragments from BmAgo3. Considering that Spn-E-EQ forms large aggregates with BmAgo3 (Fig. 1C), we speculate that Spn-E uses its ATPase activity to dynamically regulate the dissociation of protein(s) and/or remodeling of the BmAgo3 complex on target RNAs, thereby allowing the handover of BmAgo3-cleaved fragments to Siwi. This action of Spn-E mirrors the previously discussed role of Vasa in the Siwi-to-BmAgo3 handover (Xiol et al, 2014). In flies, Spn-E has been implicated in the ping-pong cycle based on its germline-restricted expression and requirement for germline piRNA production (Vagin et al, 2006; Lim and Kai, 2007; Malone et al, 2009), but its exact site of action has not been determined. Since Siwi and BmAgo3 are orthologs of Aub and Ago3, respectively, fly Spn-E could also function in the heterotypic ping-pong from Ago3 to Aub. In our current study, we knocked down Spn-E only transiently, but a continued reduction in A → S piRNAs should inevitably lead to a decrease in S → A piRNAs in the ping-pong cycle, ultimately causing a collapse of the entire piRNA pathway. This could explain the sterile phenotype of *spn-E* mutants in flies and silkworms (Gillespie and Berg, 1995; Stapleton et al, 2001; Chen et al, 2023).

Another ATP-dependent helicase, DDX43, was previously reported to be the helicase responsible for the release of cleavage fragments from BmAgo3 (Murakami et al, 2021). Indeed, we were able to confirm the reported activity of the recombinant DDX43 protein in vitro (Fig. EV1I). However, we found no evidence to support the requirement of DDX43 for A → S piRNA production in cells (Figs. 3B,D and EV2H). Further investigation is needed to clarify the biological role of DDX43.

Unexpectedly, we found that Spn-E KD hyperactivates Siwi–Siwi homotypic ping-pong and that ATPase-deficient Spn-E-EQ can rescue this phenotype (Fig. 2D,E). Thus, the presence of Spn-E itself suppresses the homotypic ping-pong of Siwi independent of its ATPase activity. A similar increase in the Aub–Aub homotypic ping-pong has been reported in *qin* mutant flies (Zhang et al, 2011). Considering that Spn-E forms a complex with Qin and Mael and binds to unloaded Siwi (Nishida et al, 2015; Namba et al, 2022), their binding to empty Siwi may repress Siwi–Siwi homotypic ping-pong. Alternatively, but not mutually exclusively, the observed increase in S → S piRNAs by Spn-E KD may be due to an excess of Siwi proteins that cannot participate in heterotypic ping-pong. In either case, the dual role of Spn-E is critical for supporting the heterotypic ping-pong amplification of piRNAs. On the other hand, our observations are based on a transient knockdown of Spn-E and overexpression of its ATPase mutant, and it is currently unknown whether piRNAs can be produced by the Siwi–Siwi homotypic ping-pong in a complete loss of Spn-E. Of note, our analysis also revealed that the homotypic Siwi–Siwi ping-pong occurs to some extent at the basal level in naive BmN4 cells (Fig. EV2G), suggesting that it may have a biological role. Further research is required to understand the mechanistic and functional differences between the heterotypic Siwi-BmAgo3 ping-pong and the homotypic Siwi–Siwi ping-pong.

## Methods

### Cell culture, plasmid transfection, and dsRNA transfection in BmN4 cells

BmN4 cells derived from silkworm ovaries were cultured in IPL-41 medium (AppliChem and HyClone) supplemented with 10% fetal bovine serum at 27 °C. Sf9 cells were cultured in Sf-900™ II SFM (Thermo Fisher Scientific/Gibco) at 28 °C on an orbital shaker set at 135 rpm. For immunoprecipitation experiments, a total of 4.5–6 µg of plasmid and dsRNA were transfected into BmN4 cells ($\sim 2 \times 10^6$ cells per 10-cm dish) with X-tremeGENE HP DNA Transfection Reagent (Merck Millipore/Roche). The second transfection was performed 3 days after the first transfection and the cells were harvested after an additional 5 days. For immunofluorescence experiments, a total of 0.4–0.5 µg of plasmid and dsRNA were transfected into BmN4 cells ($4$–$6 \times 10^4$ cells per glass bottom 35 mm dish) with X-tremeGENE HP DNA Transfection

Reagent (Merck Millipore/Roche), and the cells were fixed 5–6 days later. For artificial piRNA reporter experiments, a total of 3–5 µg of plasmid and dsRNA were transfected into BmN4 cells ($\sim 2 \times 10^6$ cells per 10-cm dish) with X-tremeGENE HP DNA Transfection Reagent (Merck Millipore/Roche). Transfection was repeated 2 days and 5 days after the first transfection and the cells were harvested after an additional 4 days. For library preparation, a total of 5 µg of plasmid and dsRNA were transfected into BmN4 cells ($8 \times 10^5$ cells per 10-cm dish) with X-tremeGENE HP DNA Transfection Reagent (Merck Millipore/Roche) every 3 days four times. dsRNA preparation was described previously (Izumi et al, 2020). Template DNAs were prepared by PCR using primers containing the T7 promoter, as listed in Table EV1.

## Plasmid construction

pIExZ-FLAG-Spn-E was described previously (Chung et al, 2021). pIExZ-FLAG-Spn-E-E251Q was generated by site-directed mutagenesis.

### pIEx-FLAG-DDX43, pIExZ-HA-DDX43

A FLAG-tagged or HA-tagged cDNA fragment of DDX43 was amplified by RT-PCR from total RNA extracted from silkworm ovaries and cloned into the pIEx-1 vector (Merck Millipore/Novagen) or the pIExZ vector using an In-Fusion cloning kit (Takara).

### pCold-DDX43, pCold-DDX6

A DNA fragment encoding DDX43 or DDX6 was amplified by PCR and inserted into the pCold-I vector (Takara) using an In-Fusion cloning kit (Takara).

### pFastBac-6HFLAGSBP-Spn-E-WT, E251Q

A DNA fragment coding Spn-E-WT or E251Q was amplified by PCR and subcloned into the pcDNA5/FRT/TO vector (Thermo Fisher Scientific) with the FLAGSBP sequence inserted. Then, a DNA fragment encoding 6HisFLAGSBP-Spn-E-WT and E251Q was amplified by PCR and inserted into the pFastBac vector (Thermo Fisher Scientific) using an In-Fusion cloning kit (Takara).

### pIEx4-piR484-A reporter

Synthesized DNA oligos with chimeric target sequences of piR484 and piR2986 were annealed (see Fig. EV3A) and inserted into the BamHI and HindIII sites of the pIEx4 vector (Merck Millipore/Novagen).

### pIB-piggyBac piRNA reporter

The *piggyBac* piRNA target site was inserted into the middle of the V5 coding sequence in the pIB-V5/His vector (Thermo Fisher Scientific) by site-directed mutagenesis. For linearization, the plasmid was digested with *AgeI*, which cleaves 20 bp downstream of the *piggyBac* piRNA target site (see Fig. 3C).

The sequences of primers used for plasmid construction are listed in Table EV1.

## Antibodies and western blotting

Rabbit anti-Siwi, anti-BmAgo3, anti-Spn-E, and biotinylated BmAgo3 antibodies were described previously (Izumi et al, 2020, 2022). A rabbit anti-DDX6 antibody was generated by immunizing N-terminally His-tagged full-length DDX6. Anti-FLAG (M2, Merck Millipore/Sigma), anti-DYKDDDDK peroxidase-conjugated (Wako), anti-HA (3F10, Merck Millipore/Roche), and anti-α-Tubulin (B-5-1-2, Merck Millipore/Sigma) antibodies were purchased. Chemiluminescence was induced by Luminata Forte Western HRP Substrate (Merck Millipore) or SuperSignal™ West Femto Maximum Sensitivity Substrate (Thermo Fisher Scientific), and images were acquired by an Amersham Imager 600 (Cytiva).

## Immunoprecipitation

For BmAgo3 immunoprecipitation, cells were resuspended in buffer A [25 mM Tris-HCl (pH 7.6), 150 mM NaCl, 1.5 mM MgCl$_2$, 0.2% sodium deoxycholate, 0.1% lithium dodecyl sulfate, 0.4% NP-40, 0.5 mM DTT, 1× Complete EDTA-free protease inhibitor (Merck Millipore/Roche), 1× PhosSTOP (Merck Millipore/Roche)] and incubated on ice for 20 min. The cell suspension was centrifuged at $17,000 \times g$ for 30 min at 4 °C, and the cleared lysate was diluted with an equal volume of buffer A without detergents. The cell lysate was incubated with normal rabbit IgG (Cell Signaling Technology) or an anti-BmAgo3 antibody at 4 °C for 1 h, and then Dynabeads Protein G (Thermo Fisher Scientific/Invitrogen) was added. After incubation at 4 °C for 1.5 h, the beads were washed five times with buffer A without protease inhibitors and phosphatase inhibitors, and the immunopurified complex was eluted with SDS sample buffer. For FLAG-Spn-E immunoprecipitation, cells were resuspended in buffer B [25 mM Tris-HCl (pH 7.6), 150 mM NaCl, 1.5 mM MgCl$_2$, 0.25% Triton X-100, 0.5 mM DTT, 1× Complete EDTA-free protease inhibitor (Merck Millipore/Roche), 1× PhosSTOP (Merck Millipore/Roche)] and homogenized with a Dounce homogenizer on ice. The cell lysate was centrifuged at $17,000 \times g$ for 30 min at 4 °C, and the supernatant was incubated with Dynabeads Protein G (Thermo Fisher Scientific/Invitrogen) pre-conjugated with an anti-FLAG antibody (M2; Merck Millipore/Sigma) at 4 °C for 1.5 h. The beads were washed five times with buffer C [25 mM Tris-HCl (pH 7.4), 150 mM NaCl, 1.5 mM MgCl$_2$, 0.5% Triton X-100, 0.5 mM DTT], and the immunopurified complexes were eluted with 3× FLAG peptide (Merck Millipore/Sigma). For the tandem IP experiment, the eluate was diluted with an equal volume of buffer D [30 mM HEPES–KOH (pH 7.4), 100 mM KOAc, 2 mM Mg(OAc)$_2$, 0.5 mM DTT]. One-third of the lysate was incubated with normal rabbit IgG (Cell Signaling Technology) and the remaining was incubated with an anti-BmAgo3 antibody at 4 °C for 30 min, and then Dynabeads Protein G (Thermo Fisher Scientific/Invitrogen) was added. After incubation at 4 °C for 1.5 h, the beads were washed five times with buffer C. The BmAgo3 immunoprecipitated beads were divided into two and incubated in buffer D with or without 200 µg/ml RNase A (Qiagen) at 30 °C for 15 min. The beads were washed twice with buffer D, and the immunopurified complex was eluted with SDS sample buffer. For Siwi immunoprecipitation in artificial piRNA reporter experiments, cells were resuspended in buffer B and homogenized with a Dounce homogenizer on ice. The cell lysate was centrifuged at $17,000 \times g$ for 30 min at 4 °C. The supernatant was further supplemented with Triton X-100 (final concentration, 1%) and incubated with an anti-Siwi antibody at 4 °C for 1 h, and then Dynabeads Protein G (Thermo Fisher

Scientific/Invitrogen) was added. After incubation at 4 °C for 1.5 h, the beads were washed five times with buffer E [25 mM Tris-HCl (pH 7.4), 150 mM NaCl, 1.5 mM MgCl$_2$, 1% Triton X-100, 0.5 mM DTT]. A portion of the immunopurified complexes was eluted with SDS sample buffer, and the remainder was eluted with TRI Reagent (Molecular Research Center) for northern blot analysis.

## Purification of recombinant proteins

pCold-DDX43 and pCold-DDX6 were transformed into Rosetta 2 (DE3) competent cells (Merck Millipore/Novagen). The cells were cultured at 37 °C until the OD$_{600}$ reached ~0.6, and then cooled on ice for 30 min. Protein expression was induced with 1 mM IPTG at 15 °C overnight. The cell pellets were resuspended in His purification buffer [50 mM HEPES–KOH (pH 7.4), 300 mM NaCl, 13 mM imidazole, 0.2 mM TECP, 10 μg/ml leupeptin, 10 μg/ml aprotinin, 1 μg/ml pepstatin A, 1× Complete EDTA-free protease inhibitor cocktail (Merck Millipore/Roche)] and sonicated with Bioruptor II (CosmoBio). The cell lysate was centrifuged at 17,000 × $g$ at 4 °C for 20 min. The cleared lysate was added to cOmplete His-Tag Purification Resin (Merck Millipore/Roche) and incubated at 4 °C for 1.5 h. After the resin was washed with the His purification buffer, the bound proteins were eluted with elution buffer [50 mM HEPES–KOH (pH 7.4), 150 mM NaCl, 300 mM imidazole, 0.2 mM TECP]. The peak fractions were pooled and dialyzed with PBS overnight. 6HisFLAGSBP-Spn-E (WT or E251Q) was expressed in SF9 cells by the Bac-to-Bac Baculovirus Expression system (Thermo Fisher Scientific/Invitrogen). The SF9 cells were resuspended in buffer F [25 mM Tris-HCl (pH 7.4), 150 mM NaCl, 1.5 mM MgCl$_2$, 0.5% Triton X-100, 0.4% NP-40, 1 mM DTT, 1× Complete EDTA-free protease inhibitor (Merck Millipore/Roche), 1× PhosSTOP (Merck Millipore/Roche)] and homogenized with a Dounce homogenizer. The cell homogenate was centrifuged at 17,000 × $g$ for 30 min at 4 °C. The supernatant was supplemented with Triton X-100 (1%, final concentration), NaCl (450 mM, final concentration), and RNase A (150 μg/ml, final concentration; Qiagen) and incubated with Streptavidin Sepharose High Performance (Cytiva) at 4 °C for 2 h. The Sepharose beads were washed with buffer G [25 mM Tris-HCl (pH 7.4), 550 mM NaCl, 1.5 mM MgCl$_2$, 1% Triton X-100, 0.4% NP-40, 1 mM DTT] and rinsed with buffer H [25 mM Tris-HCl (pH 7.4), 150 mM NaCl, 1.5 mM MgCl$_2$, 0.05% Triton X-100, 1 mM DTT]. The 6HisFLAGSBP-Spn-E (WT or E251Q) protein was eluted with buffer H containing 2.5 mM biotin, and the eluate was dialyzed with PBS overnight.

## In vitro cleavage fragment release assay

To generate 5′ and internally radiolabeled target RNAs, synthesized 5′ and 3′ fragments of the target RNA were $^{32}$P-radiolabeled at the 5′ end using T4 polynucleotide kinase (Takara). After gel purification, the radiolabeled 5′ fragment was ligated to a 5′-phosphorylated unlabeled 3′ fragment, and the radiolabeled 3′ fragment was ligated to an unlabeled 5′ fragment by splinted ligation with T4 DNA ligase (NEB) at 30 °C for 2 h, respectively. The radiolabeled target RNA was gel-purified and used for the assay. BmAgo3 immunoprecipitation for the cleavage assay was described previously (Izumi et al, 2022). A target cleavage assay was

performed at 40 °C for 2 h in a 10 μl reaction containing 3 μl of 40× reaction mix (Haley et al, 2003) and 2 nM $^{32}$P-radiolabeled target RNA. After the supernatant was removed, the BmAgo3-bound beads were further incubated in buffer D containing 5 mM ATP and 350 nM recombinant proteins at 30 °C for 1 h. Then, the supernatant and bead fractions were treated separately with proteinase K, and the target RNA was purified by EtOH precipitation. An image of the target RNA, separated on an 8% denaturing polyacrylamide gel, was captured using an FLA-7000 imaging system (Fujifilm Life Sciences). The oligonucleotides used for target RNA preparation are listed in Table EV1.

## ATPase assay and thin-layer chromatography

ATP hydrolysis reaction was performed at 25 °C for 1 h using 2 μg of recombinant protein in buffer D containing 0.02% Triton X-100, 2 μM 36-nt polyU RNAs, and 0.23 μl of [γ-$^{32}$P] ATP (6000 Ci/mmol, Perkin Elmer). Twenty percent of the reaction mixture was spotted onto a polyethyleneimine cellulose plate (MACHEREY-NAGEL) that had been pre-run with water for 2 h. The plate was then run with 450 mM ammonium sulfate for 1 h, dried, and analyzed using an FLA-7000 imaging system (Fujifilm Life Sciences).

## RNA extraction, northern blotting, and quantitative real-time PCR

For real-time PCR, northern blotting, and preparation of small RNA libraries, total RNA was prepared using TRI Reagent (Molecular Research Center) or the mirVana miRNA Isolation Kit (Thermo Fisher Scientific/Invitrogen). Northern blotting and quantitative real-time PCR were performed as described previously (Izumi et al, 2020). The probes for northern blotting and the primer sequences for real-time PCR are listed in Table EV1.

## Immunofluorescence

BmN4 cells were fixed with 4% paraformaldehyde in PBS at room temperature for 10 min, then the cells were permeabilized with 0.3% Triton X-100 in PBS for 5 min. After pre-incubation with blocking buffer [PBS supplemented with 1% BSA (Merck Millipore/Sigma) and 0.1% Triton X-100] at room temperature for 1 h, the cells were incubated with primary antibodies [anti-FLAG antibody (M2; Merck Millipore/Sigma, 1/300), anti-HA antibody (3F10; Merck Millipore/Roche, 1/300), anti-Siwi antibody (1/400), anti-BmAgo3 antibody (1/300), anti-Spn-E antibody (1/250), or anti-DDX6 antibody (1/250)] in blocking buffer at 4 °C overnight. Alexa Fluor 488 donkey anti-mouse IgG, Alexa Fluor 488 donkey anti-rat IgG, Alexa Fluor 647 donkey anti-rabbit IgG, Alexa Fluor 647 goat anti-rat IgG, Dylight 488 goat anti-rabbit IgG, and Dylight 594 goat anti-mouse IgG secondary antibodies (Thermo Fisher Scientific) were used for detection. A biotinylated anti-BmAgo3 antibody was used after incubation with the secondary antibody, and the BmAgo3 signal was detected by Cy3-Streptavidin (Jackson ImmunoResearch). Images were captured using an Olympus FV3000 confocal laser scanning system with a ×60 oil immersion objective lens (PLAPON 60XO, NA 1.42; Olympus) and processed with FV31S-SW Viewer software and Adobe Photoshop Elements 10.

## Mass spectrometry analysis

Immunoprecipitated BmAgo3 complexes separated by SDS-PAGE were stained with Coomassie brilliant blue (CBB), and the p160 band was excised from the gel, followed by in-gel digestion with trypsin (Promega) at 37 °C for overnight. LC-MS/MS analysis was conducted using an LTQ-Orbitrap Velos mass spectrometer (Thermo Fisher Scientific) equipped with a nanoLC interface (Zaplous Advance nanoUHPLC HTS-PAL xt System) (AMR). The nanoLC gradient was delivered at 500 nL/min and consisted of a linear gradient of mobile phase developed from 5 to 45% of acetonitrile in 60–180 min. Proteins were identified by the search algorithm Proteome Discoverer 2.4 (Thermo Fisher Scientific) using the protein database of *Bombyx mori* from NCBI.

## Small RNA library preparation

Small RNA libraries were constructed from 20 to 50 nt total RNA according to the Zamore laboratory's open protocol (https://www.dropbox.com/s/r5d7aj3hhyaborq/) with some modifications (Fu et al, 2018). The 3′ adapter was conjugated with an amino CA linker instead of dCC at the 3′ end (GeneDesign) and adenylated using a 5′ DNA adenylation kit at the 5′ end (NEB). To reduce ligation bias, four random nucleotides were included in the 3′ and 5′ adapters [(5′-rAppNNNNTGGAATTCTCGGGTGCCAAGG/ amino CA linker-3′) and (5′-GUUCAGAGUUCUACAGUCC-GACGAUCNNNN-3′)] and adapter ligation was performed in the presence of 20% PEG-8000. After 3′ adapter ligation at 16 °C for ≥ 16 h, the RNAs were size-selected by urea PAGE. For RNA extraction from a polyacrylamide gel, a ZR small RNA PAGE Recovery Kit (ZYMO Research) was used. Small RNA libraries were sequenced on a HiSeq 4000 or DNBSEQ-G400 platform.

## Sequence analysis of small RNA libraries

First, the adapter was removed by cutadapt using the sequence of the common part of the 3′ adapter sequence after NNNN (Martin, 2011). The sequences were subsequently converted to fasta using fastq_to_fasta in the FASTX-Toolkit, and the completely duplicated sequences were removed using fastx_collapser to thin out to one (FASTX-Toolkit; http://hannonlab.cshl.edu/fastx_toolkit/). Finally, Cutadapt was used to remove 4 nts of the UMI sequence from the 5′ and 3′ ends. Final sequences longer than 12 nt were used for analysis. Libraries were normalized by the number of reads that mapped to the genome allowing a single-nucleotide mismatch by bowtie.

Mapping to transposons was performed using bowtie, allowing for a single-nucleotide mismatch (Langmead et al, 2009). SAM files were converted to bam files by SAMtools (Li et al, 2009) and then to bed files by BEDTools (Quinlan and Hall, 2010). Among the mapped reads, reads between 23 and 32 nts in length were extracted as piRNAs using the awk command in Linux. The number of reads mapped to each transposon was measured using coverageBed in BEDtools, and the results were read into R to create a MA plot (Quinlan and Hall, 2010).

For individual piRNAs, mapping was performed with bowtie, allowing single-nucleotide mismatches and multiple mapping to the surrounding regions of 3236 previously constructed piRNA sequences (Izumi et al, 2016). SAM files were converted to bam files by SAMtools (Li et al, 2009) and then to bed files by BEDTools (Quinlan and Hall, 2010). Among the mapped reads, reads between 23 and 32 nts were extracted as piRNAs using the awk command in Linux. The results were used to obtain the position of the 5′ end of the nearby piRNA for each piRNA using a custom script in R for subsequent analysis. Of the 3236 piRNAs, 352 derived from TE1_bm_1645_LINE/R4, a transposon containing rRNA, were excluded from the analysis.

### Definition of S→A, A→S, and S→S piRNAs

Only small RNA reads with 5′-end matches to the previously defined 3236 piRNAs were extracted from the Siwi-IP and BmAgo3-IP libraries, and the IP libraries were normalized by the total read count of the 3236 piRNAs. Based on the RPM values, these piRNAs were then classified into Siwi-bound piRNAs and BmAgo3-bound piRNAs (i.e., "S" or "A" after the arrow) depending on whether they were more abundant in the Siwi-IP or the BmAgo3-IP library. Next, a similar analysis was performed on piRNAs with 5′ 10-nt overlapping sequences to determine whether the ping-ping partner was Siwi or BmAgo3 (i.e., "S" or "A" before the arrow) based on their abundance in the IP libraries. The sequences of 352 rRNA-derived piRNAs, 212 piRNAs for which the PIWI protein bound to 5′ 10-nt complementary piRNAs could not be determined (including many piRNAs that had no more than 1 read in the IP libraries), and 32 BmAgo3-bound piRNAs abundant in both the sense and antisense strands were excluded. Finally, 571 S → S piRNAs, 1036 A → S piRNAs, and 1033 S → A piRNAs were used for analysis.

## Data availability

The sequencing data are deposited in the DDBJ database under accession numbers DRA017492, DRA017493.

## Peer review information

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

## Acknowledgements

Illumina sequencing was performed by the Vincent J. Coates Genomics Sequencing Laboratory at UC Berkeley, supported by an NIH S10 OD018174 Instrumentation Grant. We thank all the members of the Tomari laboratory for critical comments on the manuscript. This work was supported in part by a Grant-in-Aid for Scientific Research (S) (grant 18H05271 to YT), a Grant-in-Aid for Scientific Research (A) (grant 23H00364 to YT), a Grant-in-Aid for Scientific Research (C) (grant 23K05632 to NI) and a Grant-in-Aid for Early-Career Scientists (grant 22K15082 to KS).

## Author contributions

**Natsuko Izumi**: Conceptualization; Funding acquisition; Validation; Investigation; Visualization; Writing—original draft. **Keisuke Shoji**: Data curation; Formal analysis; Funding acquisition; Investigation; Visualization; Methodology; Writing—review and editing. **Lumi Negishi**: Investigation. **Yukihide Tomari**: Conceptualization; Supervision; Funding acquisition; Project administration; Writing—review and editing.

## Disclosure and competing interests statement

The authors declare no competing interests.

# Expanded View Figures

**Figure EV1.   Characterization of Spn-E-EQ aggregates and BmAgo3-cleavage fragment release assay.**                                                                ▶

(**A**) Quantification of Siwi and BmAgo3 signals co-immunoprecipitated with FLAG-Spn-E (wild type or EQ) in Fig. 1D. Relative co-immunoprecipitated levels normalized to the Spn-E IP level are shown. Data are mean ± s.d. of three independent experiments (biological replicates). Statistical analysis was performed using a two-sided Student's paired *t*-test, with *p*-values adjusted using the Holm method. NS not significant. (**B**) Western blot analysis of whole cell lysates of BmN4 cells treated with dsRNA for *Rluc* (control) or *DDX6*. The anti-DDX6 antibody successfully detected endogenous DDX6. (**C**) Subcellular localization of FLAG-Spn-E-EQ, DDX6, a P-body marker protein, and BmAgo3 in BmN4 cells. Scale bar, 5 µm. (**D**) Subcellular localization of FLAG-Spn-E-EQ, BmAgo3 and Siwi in BmN4 cells. Scale bar, 5 µm. (**E**) Tandem IP experiment on BmN4 cells co-transfected with the FLAG-Spn-E-EQ expression plasmid and dsRNA targeting the *Spn-E* 3′ UTR. FLAG-Spn-E-EQ was first immunoprecipitated with the FLAG tag, and the resulting immunopurified complex was then subjected to a second IP with normal rabbit IgG (Contl-IgG) or an anti-BmAgo3 antibody. Siwi in the second immunoprecipitate was detected by western blotting with or without RNase A treatment. (**F**) Quantification of mature piRNA signals in Fig. 1E. Relative expression levels normalized to those under Spn-E-WT expression are shown. Data are mean ± s.d. of four independent experiments (biological replicates). Statistical analysis was performed using a two-sided Student's paired *t*-test, with *p*-values adjusted using the Holm method (*$p = 0.023$). (**G**) Quantification of pre-piRNA signals in Fig. 1E. Relative expression levels normalized to those under Spn-E-WT expression are shown. Data are mean ± s.d. of four independent experiments (biological replicates). Statistical analysis was performed using a two-sided Student's paired *t*-test, with *p*-values adjusted using the Holm method (*$p = 0.025$, **$p = 0.003$). (**H**) CBB staining of purified recombinant Spn-E and DDX43 proteins (rSpn-E and rDDX43, indicated by arrowheads) used for the in vitro cleavage fragment release assay. (**I**) Top: Schematic representation of the in vitro cleavage fragment release assay. To detect cleaved 5′ and 3′ fragments separately, the target RNA was radiolabeled at different positions (*). The 5′ or internally radiolabeled target RNAs were subjected to a cleavage assay using BmAgo3 immunoprecipitates. After the reaction, the bead fraction was incubated with rSpn-E or rDDX43 in the presence of ATP. Bottom: The cleaved fragments in the supernatant and bead fractions were detected by autoradiography. (**J**) Thin-layer chromatography for the detection of ATPase activity of the recombinant Spn-E protein. (**K**) Subcellular localization of FLAG-DDX43 (wild type or DA, an ATPase-deficient mutant) and BmAgo3 in BmN4 cells. Scale bar, 5 µm. (**L**) Subcellular localization of FLAG-Spn-E-EQ, HA-DDX43, and BmAgo3 in BmN4 cells. Scale bar, 5 µm.

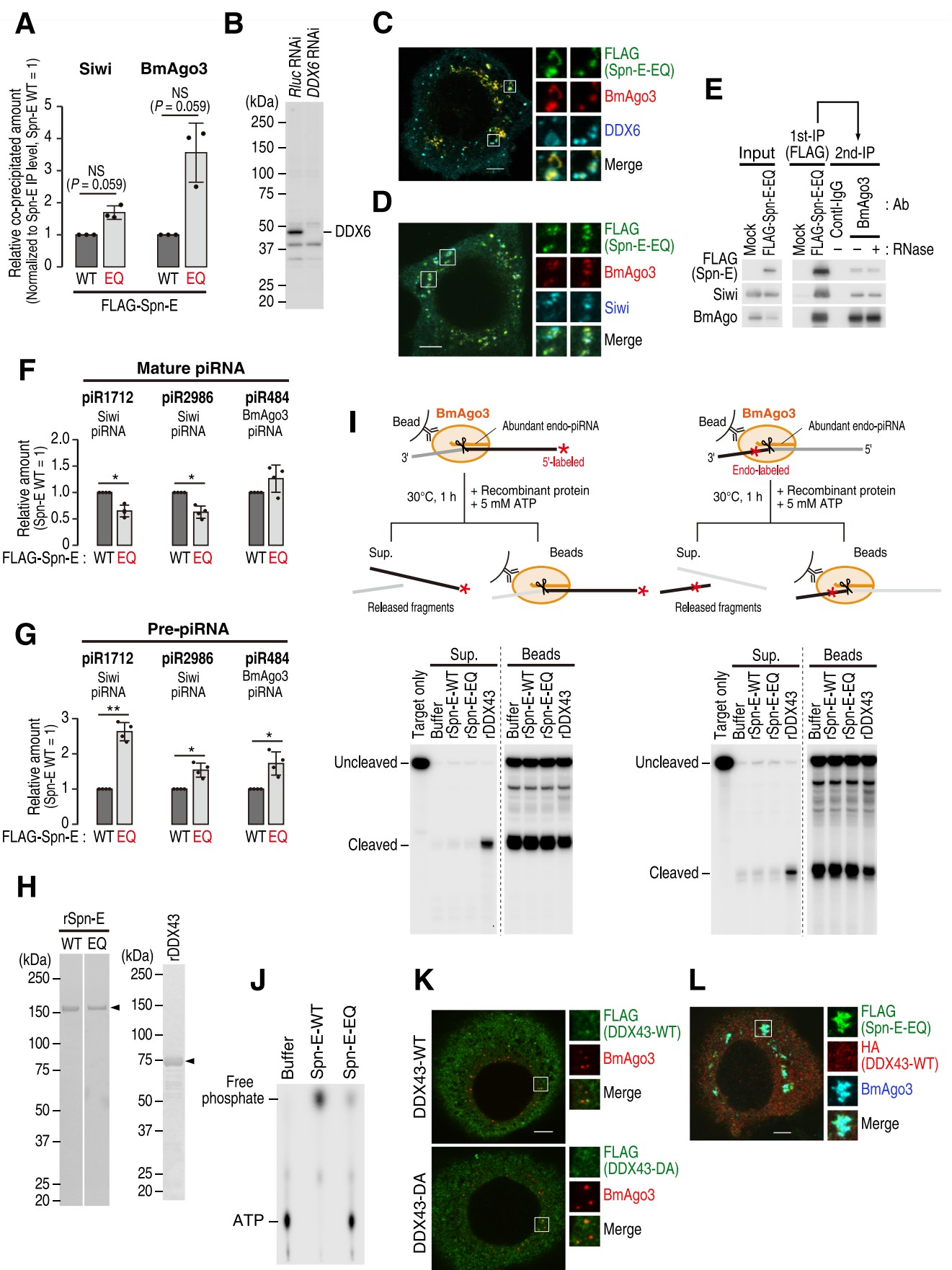

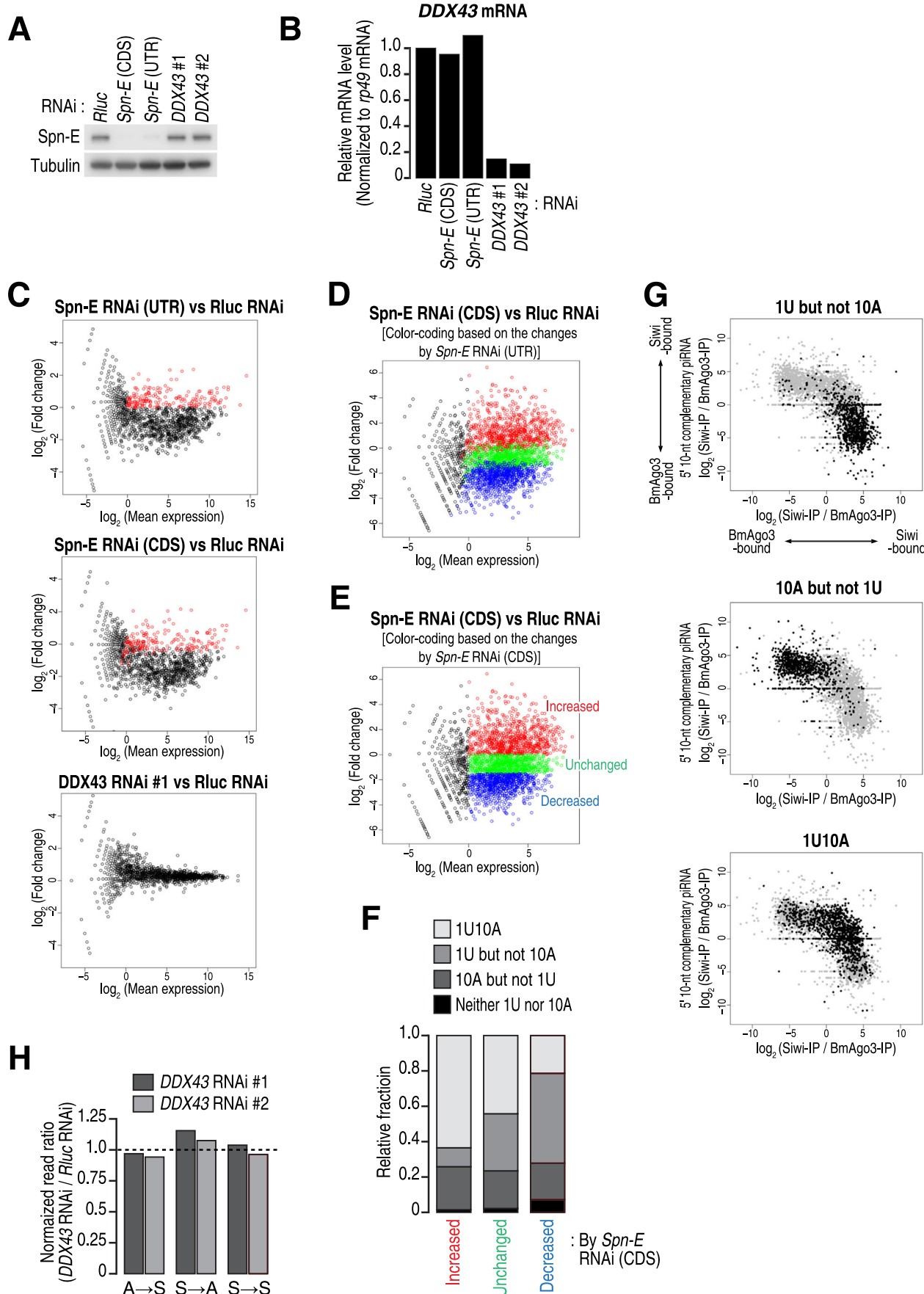

**Figure EV2. Changes in the expression of TE-mapped piRNAs upon KD of *Spn-E* or *DDX43*.**

(A) Western blot analysis of whole cell lysates from BmN4 cells treated with dsRNA for *Rluc* (control), *Spn-E*, or *DDX43*. Two different dsRNAs (*Spn-E*: CDS and 3′ UTR; *DDX43*: two different regions of the CDS, #1 and #2) were used for RNAi. *Rluc*; *Renilla luciferase*. (B) Quantitative real-time PCR analysis of the expression of *DDX43* in BmN4 cells treated with dsRNA for *Rluc* (control), *Spn-E*, or *DDX43*. Relative mRNA expression levels normalized to those of *rp49* are shown. *Rluc*; *Renilla luciferase*. (C) MA plots showing piRNA expression changes for each TE between the control KD (*Rluc* RNAi) and *Spn-E* or *DDX43* KD. Two different dsRNAs (CDS and 3′ UTR) were used for *Spn-E* RNAi. Each dot represents one TE. TEs with increased piRNA production in the *Spn-E* RNAi (UTR) are colored red. (D) MA plot showing piRNA expression changes between the control KD (*Rluc* RNAi) and *Spn-E* KD using dsRNA targeting the *Spn-E* CDS. Each dot represents one piRNA. Based on the three groups defined in Fig. 2A, piRNAs were color-coded as follows: "increased" (red, $n = 825$), "unchanged" (green, $n = 824$), and "decreased" (blue, $n = 825$). (E) MA plot showing piRNA expression changes between control KD (*Rluc* RNAi) and *Spn-E* KD using dsRNA targeting the *Spn-E* CDS. Each dot represents one piRNA. Based on the changes in expression, piRNAs were divided into three groups: "increased" (red, $n = 805$), "unchanged" (green, $n = 805$), and "decreased" (blue, $n = 805$). (F) Relative fractions of 1U10A, 1U but not 10A, 10A but not 1U, and neither 1U nor 10A piRNAs of each group in (E). (G) Scatter plots showing the PIWI binding bias of 1U but not 10A, 10A but not 1U, or 1U10A piRNAs (x-axis) and that of their putative partner piRNAs in the ping-pong cycle (y axis). (H) Changes in the expression of A → S, S → A, and S → S piRNAs in *DDX43* KD relative to control KD (*Rluc* RNAi). Two different dsRNAs targeting the *DDX43* CDS were used for RNAi. *Rluc*; *Renilla luciferase*.

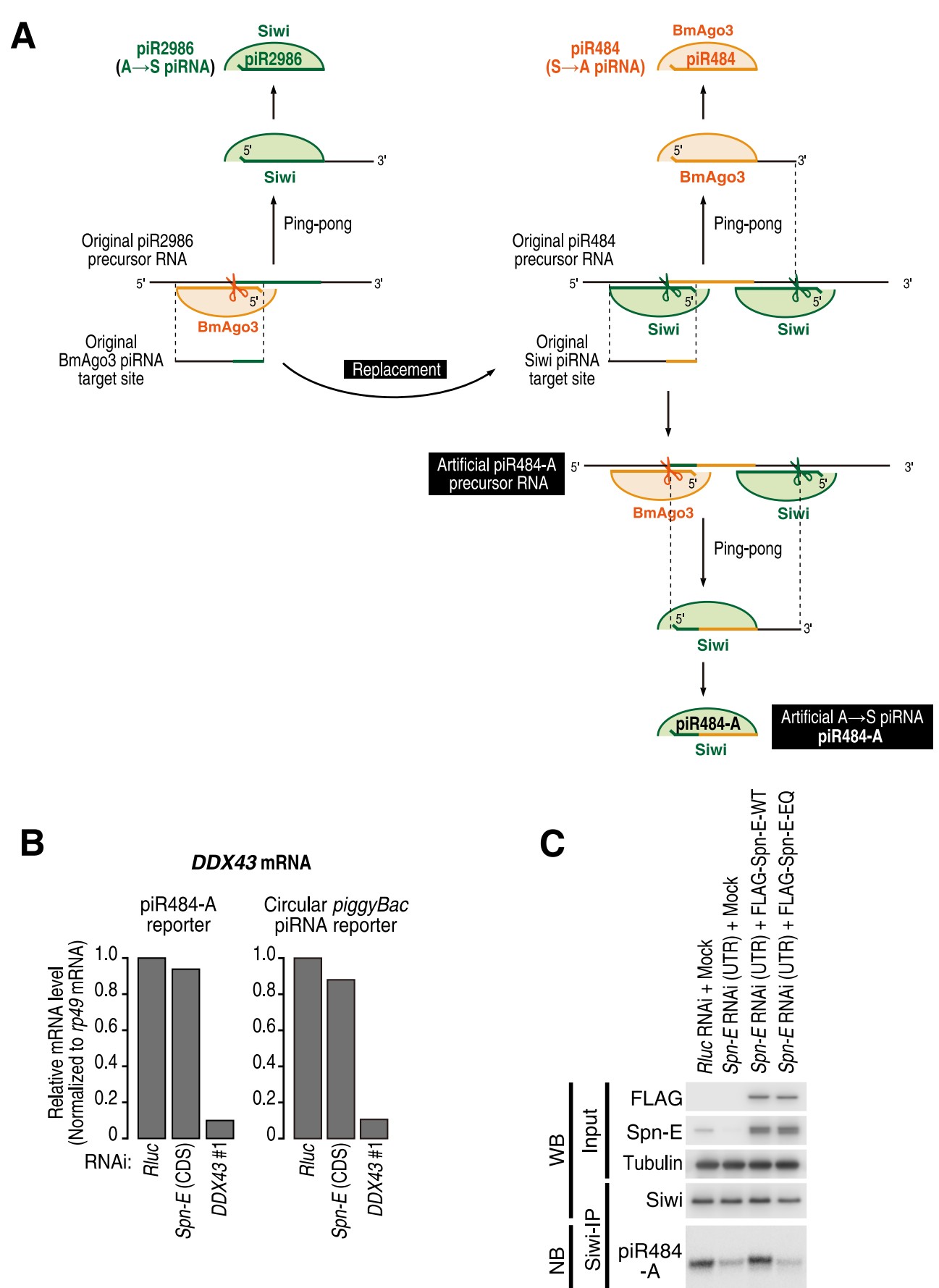

◀  **Figure EV3.  Construction of the piR484-A reporter system and Spn-E KD rescue experiment using the piR484-A reporter.**

(A) Schematic explanation of the construction of the piR484-A reporter. piR484 is an S → A piRNA, and the sequence including the 5′ 10 nt and the upstream region of piR484 was replaced with the corresponding region of piR2986, an A → S piRNA, resulting in an artificial A → S piRNA, piR484-A. The downstream sequence of piR484 that contains a Siwi-bound piRNA target sequence was used without modification. (B) Quantitative real-time PCR analysis of the expression of *DDX43* in the reporter experiments in Fig. 3B,D. Relative mRNA expression levels normalized to those of *rp49* are shown. *Rluc; Renilla luciferase*, control. (C) KD rescue experiment of Spn-E using the piR484-A reporter. Siwi was immunoprecipitated from BmN4 cells co-transfected with the piR484-A reporter plasmid, the FLAG-Spn-E expression plasmid, and dsRNA targeting the *Spn-E* 3′ UTR. Immunoprecipitated Siwi and Siwi-bound piR484-A were detected by western blotting and northern blotting, respectively. *Rluc; Renilla luciferase*, control.

