## [Peer Review File · EMBO Reports]

The dual role of Spn-E in supporting heterotypic ping-pong piRNA amplification in silkworms

Natsuko Izumi, Keisuke Shoji, Lumi Negishi, and Yukihide Tomari

Corresponding author(s): Yukihide Tomari (tomari@iqb.u-tokyo.ac.jp)

Review Timeline:

Submission Date:	19th Jan 24
Editorial Decision:	22nd Feb 24
Revision Received:	31st Mar 24
Accepted:	4th Apr 24

Editor: *Esther Schnapp*

Transaction Report:

Dear Yuki,

Thank you for the transfer of your manuscript to EMBO reports. We have now received the enclosed reports from the referees who were asked to assess it.

As you will see, both referees support the publication of your work pending only minor revisions. I am happy to say that we can therefore in principle accept it. In addition to addressing the referee concerns, a few editorial requests will need to be addressed. Please submit a point-by-point response to all requests with your final ms.

- Please reduce the number of keywords to 5.
- Please rename the conflict of interest subheading to "Disclosure Statement and Competing Interests"
- Please remove the author credits from the ms file. All credits need to be entered during online ms submission.
- Please upload with your final ms a completed author checklist that can be found here: <https://www.embopress.org/page/journal/14693178/authorguide> The completed checklist will also be part of our transparent peer-review file.
- Please also enter all funding info into our online submission system.
- Please upload all main and all EV figures as individual figure files.
- A Table EV1 is called out several times in the ms text, but not uploaded.
- Please add a direct link to the DAS that resolves to the dataset that needs to be freely accessible upon the publication of your paper.

EMBO press papers are accompanied online by A) a short (1-2 sentences) summary of the findings and their significance, B) 2-3 bullet points highlighting key results and C) a synopsis image that is exactly 550 pixels wide and 200-600 pixels high (the height is variable). You can either show a model or key data in the synopsis image. Please note that text needs to be readable at the final size. Please send us this information along with the final manuscript.

Best wishes,
Esther

Referee #1:

Review of "The dual role of Spn-E in supporting heterotypic ping-pong piRNA amplification in silkworms" by Tomari and colleagues.

The piRNA pathway is a central transposon defense system in animal gonads. Many of the key players of this pathway are highly conserved, especially those involved in piRNA processing. One of the factors with a presumed role in piRNA biogenesis, specifically in the ping-pong cycle is the RNA helicase Spn-E, a highly conserved protein in animals.

Tomari and co-workers use cultured silkworm cells that express a functional piRNA pathway centered on the ping-pong cycle and place Spn-E firmly into piRNA biogenesis. Specifically, they demonstrate a crucial function of Spn-E in the process of piRNA precursor transfer from Ago3 cleavage products to Siwi. The key findings of the work are:

- Spn-E is required for the Ago3 to Siwi piRNA biogenesis step, and for this the ATPase activity and presumably RNA helicase activity are required
- Spn-E is not required for a Siwi-Siwi homotypic ping-pong cycle; on the contrary, Spn-E protein suppresses Siwi-Siwi homotypic ping-pong in wildtype cells and for this, the ATPase activity is not required.

- Other than previously advertised, the DDX43 ATPase is not required for Ago3-Siwi ping-pong in cultured cells, even though the recombinant protein releases cleavage products from Ago3 in vitro.

The paper is extremely well written, the figures are overall a treat, and the data strongly support the author's conclusions. This work advances our understanding of a key protein involved in piRNA biogenesis, and places the obtained findings well into the literature and partially conflicting data from another group.

While the data and the analysis of the piRNA data are in itself very robust, the only critical aspect that I would like to raise is that the authors are working in a hypomorphic regime, considering that the genetics uses RNAi instead of full mutant alleles. Considering that this is probably technically very challenging and stable mutant cell lines might not be viable, the taken approach makes perfect sense. I would strongly recommend, however, to add a paragraph at the end that places the observations into this context, emphasizing which aspects of the findings might be influenced by this. For example, would an operational Siwi-Siwi ping-pong still be possible in a complete Spn-E loss of function situation? I am referring here also to the data from flies, where a full Spn-E mutant allele seemed to collapse ping-pong entirely.

Minor comments:

- Figure 1D, E: Have these experiments been performed in replicates? Can the data in 1D be quantified?
- Figure EV1C: boxes marking the zoom-windows are missing.

Referee #2:

Germline small RNAs called piRNAs are tasked with silencing of transposable elements in the animal germline genome. This is a conserved process in organisms ranging from flies to mice and human. One of the piRNA biogenesis mechanisms depends on the endonuclease (slicer) activity of one PIWI protein, which cleaves a target RNA. Subsequently, one of the two cleavage fragments is loaded into another PIWI protein, where it matures as a new piRNA. This process is termed as the Ping-pong cycle and one of the best studied and useful system for manipulation is that operating in the ovarian Silkworm cell line BmN4. The Bombyx Ping-pong cycle involves the reciprocal loading of piRNAs by PIWI proteins Siwi and Ago3. Specific RNA helicases were previously implicated in these loading events, with Ddx43 being involved in biogenesis of Siwi-bound piRNAs using a cleavage fragment provided by Ago3.

This manuscript provides convincing data to show that a conserved RNA helicase, Spn-E (Tdrd9 in mice) is the factor responsible for biogenesis of Siwi-bound piRNAs using cleavage fragments from Ago3. Reduction of Siwi in BmN4 cells results in accumulation of Spn-E in Ago3, and ectopic expression of a catalytic-dead version of Spn-E results in accumulation of Siwi piRNA precursors. Sequence analysis shows that Spn-E is responsible for promoting Siwi piRNA biogenesis mediated by Ago3 (heterotypic), and also reducing that mediated by Siwi cleavage (homotypic), with the latter being independent of the catalytic activity of Spn-E. Such an activity is not seen with the Ddx43. Interestingly, Spn-E does not have the ability to release cleavage fragments from Ago3, as reported previously for Ddx43, perhaps clarifying that both might have a role, but at distinct steps. The authors convincingly show that reduction of Spn-E, but not that of Ddx43, results in reduction of Ago3-mediated biogenesis of Siwi piRNAs using an artificial precursor template. Mechanistically how Spn-E is involved is not clear. Overall, these results identify Spn-E as the RNA helicase required for Ago3-to-Siwi arm of the Ping-pong cycle, and clarifies the involvement of Ddx43 in this process. I support its publication.

Minor comments.

- Figure 1B and 1C: please clearly mention that Spn-E shows increasing association with Ago3 in Siwi KD or SpnE-EQ conditions. It is increasing association, as there is still co-localization under normal conditions.
- Is the recombinant Spn-E functional, for example, in an ATPase assay? Just to be sure that that is not a concern, as to why it was not able to release cleavage product from Ago3.

Point-by-point response to the reviewers' comments

Referee #1:

Review of "The dual role of Spn-E in supporting heterotypic ping-pong piRNA amplification in silkworms" by Tomari and colleagues.

The piRNA pathway is a central transposon defense system in animal gonads. Many of the key players of this pathway are highly conserved, especially those involved in piRNA processing. One of the factors with a presumed role in piRNA biogenesis, specifically in the ping-pong cycle is the RNA helicase Spn-E, a highly conserved protein in animals. Tomari and co-workers use cultured silkworm cells that express a functional piRNA pathway centered on the ping-pong cycle and place Spn-E firmly into piRNA biogenesis. Specifically, they demonstrate a crucial function of Spn-E in the process of piRNA precursor transfer from Ago3 cleavage products to Siwi. The key findings of the work are:

- Spn-E is required for the Ago3 to Siwi piRNA biogenesis step, and for this the ATPase activity and presumably RNA helicase activity are required
- Spn-E is not required for a Siwi-Siwi homotypic ping-pong cycle; on the contrary, Spn-E protein suppresses Siwi-Siwi homotypic ping-pong in wildtype cells and for this, the ATPase activity is not required.
- Other than previously advertised, the DDX43 ATPase is not required for Ago3-Siwi ping-pong in cultured cells, even though the recombinant protein releases cleavage products from Ago3 in vitro. The paper is extremely well written, the figures are overall a treat, and the data strongly support the author's conclusions. This work advances our understanding of a key protein involved in piRNA biogenesis, and places the obtained findings well into the literature and partially conflicting data from another group. While the data and the analysis of the piRNA data are in itself very robust, the only critical aspect that I would like to raise is that the authors are working in a hypomorphic regime, considering that the genetics uses RNAi instead of full mutant alleles. Considering that this is probably technically very challenging and stable mutant cell lines might not be viable, the taken approach makes perfect sense. I would strongly recommend, however, to add a paragraph at the end that places the observations into this context, emphasizing which aspects of the findings might be influenced by this. For example, would an operational Siwi-Siwi ping-pong still be possible in a complete Spn-E loss of function situation? I am referring here also to the data from flies, where a full Spn-E mutant allele seemed to collapse ping-pong entirely.

We appreciate the Reviewer's positive evaluation of our manuscript and valuable comments. We completely agree with the Reviewer about our hypomorphic experimental design, and we have clarified this point in the revised manuscript (page 16): "On the other hand, our observations are based on a transient knockdown of Spn-E and overexpression of its ATPase mutant, and it is currently unknown whether piRNAs can be produced by the Siwi-Siwi homotypic ping-pong in a complete loss of Spn-E."

Minor comments:

- Figure 1D, E: Have these experiments been performed in replicates? Can the data in D be quantified?

We have repeated the Spn-E-IP experiment of Fig 1D and obtained consistent results. In general, quantification of Western blot signals is not necessarily linear, but we nonetheless quantified the signals. We confirmed that the EQ mutation of Spn-E reproductively enhances its association with BmAgo3 more than that with Siwi, although the *p*-values for the enhancing effects were slightly above 0.05 by *t*-test with Holm correction. We have added the quantification data in Fig EV1A and provided a new data that best represents the results of the three replicates in Fig 1D (all the raw data are included in Source Data).

Similarly, we have also repeated the Northern blot analysis of Fig 1E and confirmed that the expression of Spn-E-EQ (under endogenous Spn-E knockdown) accumulates Siwi pre-piRNAs while concomitantly decreasing their mature piRNAs (piR1712 and piR2986). On the other hand, the corresponding reduction of BmAgo3-bound mature piRNA was not observed, while the pre-piRNA was slightly accumulated (piR484). We have added the quantification data in Fig EV1F and 1G and provided a new data that best represents the results of the four replicates in Fig 1E (all the raw data are included in Source Data).

- Figure EV1C: boxes marking the zoom-windows are missing.

We have added boxes to the image. Thank you for bringing this to our attention.

Referee #2:

Germline small RNAs called piRNAs are tasked with silencing of transposable elements in the animal germline genome. This is a conserved process in organisms ranging from flies to mice and human. One of the piRNA biogenesis mechanisms depends on the endonuclease (slicer) activity of one PIWI protein, which cleaves a target RNA. Subsequently, one of the two cleavage fragments is loaded into another PIWI protein, where it matures as a new piRNA. This process is termed as the Ping-pong cycle and one of the best studied and useful system for manipulation is that operating in the ovarian Silkworm cell line BmN4. The Bombyx Ping-pong cycle involves the reciprocal loading of piRNAs by PIWI proteins Siwi and Ago3. Specific RNA helicases were previously implicated in these loading events, with Ddx43 being involved in biogenesis of Siwi-bound piRNAs using a cleavage fragment provided by Ago3.

This manuscript provides convincing data to show that a conserved RNA helicase, Spn-E (Tdrd9 in mice) is the factor responsible for biogenesis of Siwi-bound piRNAs using cleavage fragments from Ago3. Reduction of Siwi in BmN4 cells results in accumulation of Spn-E in Ago3, and ectopic expression of a catalytic-dead version of Spn-E results in accumulation of Siwi piRNA precursors. Sequence analysis shows that Spn-E is responsible for promoting Siwi piRNA biogenesis mediated by Ago3

(heterotypic), and also reducing that mediated by Siwi cleavage (homotypic), with the latter being independent of the catalytic activity of Spn-E. Such an activity is not seen with the Ddx43. Interestingly, Spn-E does not have the ability to release cleavage fragments from Ago3, as reported previously for Ddx43, perhaps clarifying that both might have a role, but at distinct steps. The authors convincingly show that reduction of Spn-E, but not that of Ddx43, results in reduction of Ago3-mediated biogenesis of Siwi piRNAs using an artificial precursor template. Mechanistically how Spn-E is involved is not clear. Overall, these results identify Spn-E as the RNA helicase required for Ago3-to-Siwi arm of the Ping-pong cycle, and clarifies the involvement of Ddx43 in this process. I support its publication.

We appreciate the Reviewer's positive evaluation for our study.

Minor comments.

- Figure 1B and 1C: please clearly mention that Spn-E shows increasing association with Ago3 in Siwi KD or SpnE-EQ conditions. It is increasing association, as there is still co-localization under normal conditions.

We thank the Reviewer for this helpful suggestion. We added the following sentence to the revised manuscript (page 8). "These results indicate that Spn-E shows increased association with BmAgo3 in Siwi KD or Spn-E-EQ expression."

-Is the recombinant Spn-E functional, for example, in an ATPase assay? Just to be sure that that is not a concern, as to why it was not able to release cleavage product from Ago3.

Following the Reviewer's suggestion, we performed a thin-layer chromatography assay to confirm the ATPase activity of the recombinant Spn-E protein. We observed a clear release of a free phosphate after incubation of [γ - 32 P]-ATP with wild-type Spn-E but not with Spn-E-EQ. Therefore, the reason why we could not detect the release of cleavage products from BmAgo3 is not simply due to the lack of the activity of the recombinant Spn-E protein. We have added this data as new Fig EV1J.

Prof. Yukihide Tomari
The University of Tokyo
Institute for Quantitative Biosciences
1-1-1 Yayoi
Life Sciences Research Building B-409
Bunkyo-ku, Tokyo 113-0032
Japan

Dear Yuki,

I am very pleased to accept your manuscript for publication in the next available issue of EMBO reports. Thank you for your contribution to our journal.

Best,
Esther
